# Community-Based Rehabilitation Indicators: Validation and Preliminary Evidence for Disability in Italy

**DOI:** 10.3390/ijerph182111256

**Published:** 2021-10-26

**Authors:** Marco Tofani, Giuseppina Esposito, Anna Berardi, Giovanni Galeoto, Silvia Iorio, Maurizio Marceca

**Affiliations:** 1Department of Public Health and Infectious Diseases, Sapienza University of Rome, 00185 Rome, Italy; maurizio.marceca@uniroma1.it; 2Department of Intensive Neurorehabiiltation and Robotics, Bambino Gesù Pediatric Hospital IRCCS, 00165 Rome, Italy; 3Traumatic and Orthopedic Center A. Alesini, 00145 Rome, Italy; pinaespo@gmail.com; 4Department of Human Neurosciences, Sapienza University of Rome, 00185 Rome, Italy; anna.berardi@uniroma1.it (A.B.); giovanni.galeoto@uniroma1.it (G.G.); 5Department of Medico-Surgical Sciences and Biotechnologies, Sapienza University of Rome, 00185 Rome, Italy; silvia.iorio@uniroma1.it

**Keywords:** disability, community-based rehabilitation, community-based inclusive development, reliability, indicators

## Abstract

Community-based rehabilitation (CBR) is a multi-sectorial community strategy for guaranteeing that people with disabilities enjoy the same rights and opportunities as all other community members. CBR is organized in a five-component matrix—namely, health, education, social, livelihood, and empowerment. To measure the effectiveness of CBR, the World Health Organization (WHO) has developed standardized indicators. The objective of the present study is to translate and validate the CBR indicators (CBR-Is), providing preliminary evidence of their use for disability in Italy. After obtaining permission from the WHO, the CBR-Is followed a process of translation and cross-cultural adaptation according to international guidelines. An examination of internal consistency and reliability was than performed. The intra-rater reliability was estimated using the Intraclass Correlation Coefficient with a 95% confidence interval. In order to measures the differences between people with and without disabilities, an independent sample *t*-test was used for quantitative indicators. The Italian version of the CBR-Is (IT-CBR-Is) was administered to 234 people. The internal consistency showed a good value, with a Cronbach’s alpha coefficient of 0.862, and the intra-rater reliability analysis showed solid values for each domain (range: 0.723–0.882). Statistically significant differences between people with and without disabilities were found for each domain of the CBR matrix—namely, health, social, education, livelihood, and empowerment. The IT-CBR-Is are consistent and reliable measures when used to investigate disability in a community-based inclusive development perspective. National stakeholders can now have specific indicators to implement services and actions for people with disabilities.

## 1. Introduction

Community-based rehabilitation (CBR), also known as community-based inclusive development, is a community action to ensure that people with disabilities have the same rights and opportunities as all other community members [1]. It was initiated by the World Health Organization (WHO) following the International Conference on Primary Health Care where the Alma-Ata Declaration was approved in 1978 [2]. In 2003, an international meeting was held to define recommendations for CBR [3]. Subsequently, the International Labor Organization (ILO), the United Nations Educational, Scientific, and Cultural Organization (UNESCO), and the WHO signed a joint “position paper” to propose CBR as a strategy for rehabilitation, equalization of opportunities, poverty reduction, and social inclusion of people with disabilities [4]. In 2005, the WHO Assembly adopted a resolution for disability prevention and rehabilitation by urging Member States to promote and strengthen CBR programs [5]. CBR was finally included in the Global Disability Action Plan 2014–2021 [6]. The action plan was endorsed by WHO Member States in 2014 and calls for them to: (a) remove barriers and improve access to health services and programs; (b) strengthen and extend rehabilitation, assistive devices and support services, and community-based rehabilitation; (c) enhance collection of relevant and internationally comparable data on disability and conduct research on disability and related services. Achieving the objectives of the action plan better enables people with disabilities to fulfill their aspirations in all aspects of life [6].

To date, CBR strategies have been developed in more than 90 countries. When measuring effectiveness, qualitative approaches have taken the upper hand in CBR and remain highly relevant. However, there is also a call for the inclusion of quantitative indicators in order to capture the progress made by people participating in CBR programs [7]. Moreover, CBR has a positive and significant impact on access to services, rights, and opportunities of people with disabilities [8], and has demonstrated its efficacy in low- and middle-income countries [9]. However, the methodological constraints of many of these studies limit the strength of their results. In order to build stronger evidence, future studies will need to adopt better study designs while also focusing on broader client groups and including economic evaluations [10]. There is also a need for changes in CBR evaluation methodologies in response to the evolution of disability models from medical models to human rights models while also considering the diversity among persons with disabilities in interpreting life experiences and their quality of life [11]. Therefore, in order to support the growth of CBR worldwide, there is also a need for a strong evidence on the effectiveness of the programs [12,13].

The lack of data for supporting the effectiveness of CBR is due, in part, to the absence of standardized indicators [14]. For this reason, the WHO and IDDC CBR Task Force decided to work together to develop indicators and questions to inform them. Indicators were developed in four steps: (1) analysis of all work pertaining to CBR; (2) reprogramming of desirable CBR results contained in the CBR Guidelines; (3) creation of an Alpha Version of CBR indicators; (4) feasibility and validity testing [15]. The CBR Indicators Manual proposes a simple and flexible data collection strategy that can be customized based on the desired indicators [16]. The indicators correspond to the components of the CBR matrix (health, education, livelihood, social life, and empowerment) and each of their five sub-elements, and they have been outlined on the basis of the desirable outcomes of CBR set out in the CBR Guidelines [15].

The CBR indicators (CBR-Is) can be used to register the differences between people who live with/without a condition of disability in each domain of the CBR matrix. The CBR-Is can be used by managers, community workers, volunteers, researchers, and other stakeholders interested in the implementation of CBR. Moreover, these indicators can be used to assess the current situation and monitor the differences that CBR is making in the lives of people with disabilities in the areas where it is implemented. It is also possible to use the indicators to monitor other action plans/interventions within communities. Considering that in Italy, there are several projects with a community-based approach [17,18,19] and a need for monitoring programs from a community-based perspective, the objective of this investigation is to: (a) translate and culturally adapt the CBR-Is into Italian; (b) evaluate its reliability, particularly its internal consistency and inter-rater reliability; (c) measure the differences between people with and without disabilities in each domain of the CBR matrix.

## 2. Materials and Methods

This investigation was carried out by researchers from Sapienza University of Rome and the Rehabilitation and Outcome Measures Assessment (ROMA) Association. The research group was involved in the validation of different outcome measures, with particular interest in disabilities and global health education [20,21,22].

### 2.1. Assessment Tool

The CBR-Is [15] are composed of 40 core and supplementary indicators. The 13 core CBR-Is are divided as follows: two for health, six for education, three for livelihood, one for social, and one for the empowerment component. The core CBR-Is are able to register differences between people with and without disabilities, regardless of individual CBR programs, as well as specific activities. The use of the core CBR-Is is recommended as a minimum set to assess the effectiveness and monitor the progress of CBR programs. Instead, the remaining 27 CBR-Is can be selected based on specific community needs according to each component of the CBR matrix. For more information, please see the CBR Indicators Manual available on the WHO website [16].

### 2.2. Translation and Cross-Cultural Adaptation

After receiving permission from the WHO, the CBR-Is were translated and culturally adapted according to international guidelines. The original version was translated into Italian by two native English speakers, who produced two independent translations. An independent native Italian speaker then synthesized the results of the two translations into one document. Two Italian translators then translated that document back into the English language without having seen the original English version of the CBR-Is. The back-translated version of the instrument was then compared with the original English version. In order to adapt the translated version to Italian culture, five Italian health professionals (two medical doctors and three rehabilitation professionals)—familiar with both the English and Italian languages and with a great deal of experience with CBR—reviewed the first translated version and then reworded and reformulated some items to minimize the differences from the original version. After obtaining the pre-final version, in order to be sure about the comprehensibility of the scale, a pre-test was conducted, involving five people with and without disabilities. The equivalence of the two versions was investigated in regard to their semantic domains. The Italian version of the CBR-Is (IT-CBR-Is) was formed as a result of the translation and cross-cultural adaptation process.

### 2.3. Sampling, Procedure, and Data Analysis

Individuals were recruited from community settings from various parts of Italy between March 2018 and March 2019. The convenience sample met the following inclusion criteria: healthy people and/or people who had a condition of disability, aged 18 years or older. A convenience sample was recruited from different community settings, again from different areas of the country. Recruitment strategies included the use of email invitations to different people, acquaintances, senior centers, university students, and employees. As recommended by the Consensus-Based Standards for the Selection of Health Status Measurement Instruments (COSMIN) [23,24], a sample size of at least 100 persons was deemed adequate for internal consistency and reliability, and at least 200 people were adequate for investigating cross-cultural validity. Consequently, the research group decided to recruit a minimum sample size of 200 people.

Before starting, the research group participated in an internal training course in order to level out confidence with the outcomes and indicators. Lessons focusing on theoretical and practical activities regarding the administration of the IT-CBR-Is were also organized. Socio-demographic information was obtained from direct interviews with the participants. According to the COSMIN, in order to evaluate internal consistency, Cronbach’s alpha coefficient was used. Internal consistency measures the relatedness of the items and consistency of the scale [25]—it was calculated for the total score of the IT-CBR-Is. As reported by Nunually [26], a satisfactory index of the homogeneity of the scale should have at least an alpha of 0.70. For the reliability study, the intra-rater reliability was investigated with the intraclass correlation coefficient (ICC) with 95% confidence intervals. Intra-rater reliability was determined by the evaluation of the same individual at different moments within a week. The ICC estimates ranged from 0 (no agreement) to 1 (perfect agreement) and were interpreted as follows: 0.00–0.25, little or no correlation; 0.26–0.50, low correlation; 0.51–0.070, moderate correlation; 0.71–0.90, high correlation; 0.91–1, very high correlation [25,27].

In order to obtain preliminary evidence on how the IT-CBR-Is can capture the differences between people with and without disabilities, an independent sample *t*-test was applied for questions in which it was possible to transform nominal variables into numerical variables, as provided in the original manual produced by the WHO. Therefore, some questions of a descriptive nature were excluded (e.g., H06 and H09: “Which reason(s) explain(s) why you did not get that health/rehabilitation service?”) or other questions related to the use of assistive technologies. Significance was set to *p* < 0.05 with 95% confidence intervals. All analyses were performed by using the Statistical Package for the Social Sciences (SPSS) version 24.0 (IBM Corp. Released 2016. IBM SPSS Statistics for Windows, Version 24.0. IBM Corp., Armonk, NY, USA).

## 3. Results

The IT-CBR-Is were administered to 234 individuals: 40 people with disabilities (mean age: 42, SD: 14.26) and 194 without disabilities (mean age: 38, SD: 13.12). The sample was homogeneous for age and gender. The majority of the population were residents in central and southern Italy. The characteristics of the sample are summarized in Table 1.

Regarding the reliability study, the internal consistency measured with Cronbach’s alpha coefficient was 0.862 for the whole scale. The ICCs with 95% confidence intervals for intra-rater reliability were within the range of 0.723–0.882. The ICC values for each subscale of the IT-CBR-Is are reported in Table 2.

Regarding differences between people with and without disabilities, statistically significant differences were found in each domain of the CBR matrix: namely, three for health, four for social, three for education, two for livelihood, and three for empowerment. Table 3 describes the differences among people with and without disabilities for each domain of the IT-CBR-Is in greater depth.

## 4. Discussion

This study reports the Italian translation and cross-cultural adaptation of the CBR-Is and provides preliminary evidence on the IT-CBR-Is’ reliability and validity for use with people with and without disabilities aged 18 years or older.

The translation followed current international guidelines. During the pre-test phase, the involvement of both experts and recipients proved extremely useful. As a matter of fact, this step led to an improvement in the comprehensibility of the questions and their adaptation to the specific Italian context. For example, for question H10, “Do you use any aids to help you get around, such as a cane, crutch, or wheelchair; or to help you with self-care, such as grasping bars or a hand or arm brace?”, the expert suggested translating “hand or arm brace” as “ortesi” (orthotics) due to the fact that this wording is more technically appropriate. However, when we applied the pre-test phase with the target population, they reported that they did not understand what “ortesi” meant. After explaining the meaning and following a careful debate, the research group opted to translate with “tutori per la mano o l’arto superiore” (hand or upper limb brace) because this wording is more appropriate for non-technical staff. Furthermore, the CBR-Is can also be used with community-based rehabilitation workers or other personnel who lack specific training with medical aspects. Consequently, creating a version that was easy to understand appears to be important. Another relevant aspect to be considered is the specific adaptation for the Italian context. For example, we considered the different educational systems among countries. In the original version, for question E01 (“What is the highest level of education you have achieved, or are working to achieve?), different answers were provided. However, these responses were not appropriate for the Italian school system due to the fact that in Italy, there is no college. In order to solve this challenge, experts were consulted—while also gaining an understanding of the target population’s point of view—and the research group contacted the WHO headquarters; by working together, they found a solid solution. The final manual of the IT-CBR-Is was finally approved by the authors and the WHO [28].

The IT-CBR-Is revealed good internal consistency (Cronbach’s alpha coefficient of 0.862), meaning that that all items were related to each other and that they positively contributed to measuring the same general construct. For the intra-rater reliability, the IT-CBR-Is showed good stability over time (within a week), especially for the social component (0.882) and empowerment (0.796) components. These encouraging results provide preliminary evidence for using the IT-CBR-Is and monitoring action plans at different levels. However, the lack of other validation studies does not allow for a comparison with other countries.

Interesting topics of the present investigation were the differences between people with and without disabilities for each domain of the CBR. First and foremost, the health component revealed that people with disabilities experienced poor health outcomes (*p* < 0.0001) and barriers to gaining access to health care (*p* < 0.0001) and rehabilitation services (*p* < 0.0001) compared to people without disabilities. This finding is in line with the World Report on Disability [13] and the recent report on access to health-care services for people with disabilities [29]. Secondly, the education component highlighted the difficulties of people with disabilities in obtaining higher education in comparison with people without disabilities (*p* < 0.0001), as well as in attending formal and informal training for skill development (*p* = 0.016) (please note that according to the manual, in this case, the most penalizing value was the highest one). Moreover, this finding is in line with those of the World Report on Disability [13] and several studies [30,31]. For livelihood, people with disabilities were more dependent on the use of their own money (*p* < 0.0001), and they did not know how to obtain social assistance, such as for loss of income through old age, sickness, or disability (*p* < 0.0001). Furthermore, regarding the social component, significant differences were found. People with disabilities felt that they were treated with little respect and that they were not considered equal to others (*p* = 0.004); they could not make decisions about their personal relationships (*p* < 0.0001) or participate in cultural activities (*p* < 0.0001) or leisure activities (*p* < 0.0001). The findings show that a focus on social inclusion in the labor market is lacking, and the main barriers identified were related to financial factors, attitudes, health issues, and unemployment [32]. However, studies suggest that higher physical and leisure/recreation activities are associated with better quality of life [33]. Lastly, the empowerment component revealed that people with disabilities lack the necessary independence in order to make the big decisions in their lives (*p* < 0.0001), influence the community where they live (*p* = 0.012), and persuade people of their views and interests (*p* < 0.0001).

The results of this study initially confirm preliminary evidence of the indicators in recording the differences between people with and without disabilities in different areas of their lives. Despite the fact that Italy ratified the United Nations’ Convention on the Rights of Persons with Disabilities, there are still many challenges to be faced in order to achieve equity in health and access to services and to promote active participation in their social and political lives while also guaranteeing the possibility of enjoying the same rights as other people.

Despite these encouraging results, the present study has some limitations. Firstly, the relatively small sample size does not allow for a generalization of the results. Moreover, using a convenience sample may have affected the ability to reach some people who were likely to have greater needs. There are no other validation studies of the CBR-Is, and consequently, this does not allow for a comparison of our findings with similar studies. Lastly, the study did not include children with and without disability. Further studies should consider these aspects.

## 5. Conclusions

In conclusion, this investigation shows the consistency and reliability of the IT-CBR-Is as tools for measuring differences between people with and without disabilities. Consequently, Italian healthcare professionals and policymakers, as well as government at the local and national levels, can now measure the impact of their actions along with the effectiveness of their interventions from a community-based inclusive development perspective.

## Figures and Tables

**Table 1 ijerph-18-11256-t001:** Sample characteristics (sample: *n* = 234).

	People with Disabilities	People without Disabilities	*t*-Student
Age mean (SD)	42 (14.26)	38 (13.12)	0.070
**Gender**			
Female	16	64	
Male	24	130	
Total	40	194	0.396
**Regional Location**			
Northern Italy	0	12	
Central Italy	33	78	
Southern Italy	7	64	

**Table 2 ijerph-18-11256-t002:** Reproducibility of the community-based rehabilitation indicators (sample: *n* = 234).

Component	ICC	Lower Bound	Upper Bound
Health	0.744	0.694	0.789
Education	0.723	0.697	0.749
Livelihood	0.739	0.647	0.754
Social	0.882	0.859	0.903
Empowerment	0.796	0.756	0.833

ICC: Intraclass correlation coefficient.

**Table 3 ijerph-18-11256-t003:** *t*-Student analysis for independent samples (total: 234).

Questions	Without Disabilities *n* = 194 Mean (SD)	With Disabilities *n* = 40 Mean (SD)	Mean Difference	*t*	*p*
H01. In general, how would you rate your health today?	1.72 (0.64)	3.25 (0.54)	−1.528	−14.094	0.000 *
H02. On your last visit to a health-care provider, to what extent were you satisfied with the level of respect you were treated with?	3.22 (1.13)	2.90 (0.95)	0.316	1.646	0.101
E01. What is the highest level of education you have achieved, or are working to achieve?	5.54 (1.28)	4.70 (1.47)	0.836	3.646	0.000 *
L02. Do you have enough money to meet your needs?	2.79 (0.85)	2.65 (0.58)	0.144	1.020	0.309
S01. Do you feel that other people respect you? For example, do you feel that others value you as a person and listen to what you have to say?	3.26 (0.89)	2.80 (1.04)	0.458	2.871	0.004 *
M01. Do you get to make the big decisions in your life? For example, deciding who to live with, where to live, or how to spend your money?	3.98 (1.04)	2.95 (1.03)	1.029	5.667	0.000 *
H03. Has your (doctor, CBR worker, or any other health professional) ever discussed with you the benefits of eating a healthy diet, engaging in regular physical exercise, or not smoking?	1.15 (0.36)	1.05 (0.22)	0.105	1.758	0.080
H04. When was the last time you had a regular health check-up?	1.38 (0.71)	1.30 (0.96)	0.081	0.617	0.538
H05. In the last 12 months, has there been a time when you needed health care but did not get that care?	2.26 (0.73)	1.65 (0.73)	0.608	4.745	0.000 *
H07. On your last visit to a health-care provider, to what extent were you involved in making decisions for your treatment?	2.94 (1.37)	2.90 (1.00)	0.038	0.167	0.868
H08. In the last 12 months, has there been a time when you needed rehabilitation services, such as physical, occupational, or speech therapy, but did not get those services?	2.70 (0.69)	2.20 (0.75)	0.501	4.097	0.000 *
E04. Do you participate in learning opportunities to improve your skills for everyday life or work?	1.30 (0.61)	1.55 (0.54)	−251	−2.423	0.016 *
E05. To what extent does it fit your needs?	2.42 (1.44)	1.60 (1.51)	0.823	3.250	0.001 *
L03. Do you get to decide how to use your money?	4.23 (1.00)	3.05 (1.17)	1.177	6.556	0.000 *
L04. Do you know how to get financial services such as credit, insurance, grants, savings programs?	1.11 (0.62)	1.05 (0.81)	0.063	0.552	0.581
L05. Do you currently benefit from any social protection program, such as loss of income through old age, sickness, or disability?	1.48 (0.88)	1.20 (0.75)	0.285	1.887	0.060
L06. Do you know how to get social protection against loss of income resulting from old age, sickness, or disability?	1.97 (0.45)	1.55 (0.54)	0.419	5.112	0.000 *
S02. Do you get to make decisions about the personal assistance that you need (who assists you, what type of assistance, when to get assistance)?	3.47 (1.59)	3.20 (1.26)	0.274	1.025	0.306
S03. Do you get to make your own decisions about your personal relationships, such as friends and family?	4.31 (1.19)	3.25 (1.27)	1.059	5.032	0.000 *
S04. Do you get to participate in artistic, cultural, or religious activities?	3.43 (1.41)	2.15 (1.36)	1.283	5.266	0.000 *
S05. Do you get to participate in community recreational, leisure, and sports activities?	3.37 (1.38)	2.15 (1.54)	1.221	4.981	0.000 *
S06. To what extent do you know your legal rights?	2.95 (0.94)	2.65 (0.80)	0.298	1.858	0.064
S07. Do you know how to access the justice system?	1.12 (0.80)	1.20 (0.68)	−0.076	−0.559	0.577
M02. Do you think that the policies in your country provide people with disabilities equal rights to those of other people?	2.01 (0.89)	1.95 (1.08)	0.060	0.374	0.709
M03. Are you satisfied with your ability to persuade people of your views and interests?	2.90 (0.96)	2.05 (1.03)	0.847	5.010	0.000 *
M04. Do you get to influence the way your community is run?	2.01 (1.10)	1.55 (0.74)	0.460	2.524	0.012 *
M05. Did you vote in the last election?	1.05 (0.26)	1.05 (0.38)	0.002	0.031	0.975
M07. To what extent do you feel Disabled People’s Organizations adequately represent your concerns and priorities?	2.24 (1.30)	2.30 (1.10)	−0.063	−0.288	0.774

* *p* < 0.05.

## Data Availability

The data presented in this study are available on reasonable request from the corresponding author.

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
