# Peer review of "Community-Based Rehabilitation Indicators: Validation and Preliminary Evidence for Disability in Italy"

_ijerph, 2021, doi:10.3390/ijerph182111256_

Round 1

Reviewer 1 Report

Marco Tofani et.al; Community-Based Rehabilitation Indicators: Italian Validation and Preliminary Evidence for Disability, was written well. I would like to request to authors to change the following. 

  1. In abstract (range .723 - .882). it is better to write 0.723-0.882).
  2.  Reference numbering is very confusing, please check it.
  3.  consistency of notation of number:follow: two for health, 6 for education, three for livelihood, one for social, and one for empowerment component, please correct it.
  4.  table 3,  change P-value representation (eg. 0.0000)

thank you 

Author Response

Point 1. In abstract (range .723 - .882). it is better to write 0.723-0.882).

Response 1. Thank you. We modified it 

Point 2. Reference numbering is very confusing, please check it.

Response 2. Thank you for your observation. We checked and mofied 

Point 3. consistency of notation of number:follow: two for health, 6 for education, three for livelihood, one for social, and one for empowerment component, please correct it.

Response 3. Thank you. We corrected it

Point 4.  table 3,  change P-value representation (eg. 0.0000)

Response 4. Thank you, we modified

Reviewer 2 Report

I think that a more detailed explanation of the CBR program and its strategy should be included in the Intro of the paper. 

There are some grammatical mistakes in the paper. For instance, in the second paragraph of the Discussion, the word "adapt" should be "adaptation of" in the sentence, "As a matter of fact, this step led to an improvement in the comprehensibility of the questions and adapt them to the specific Italian context." Also, the word "fundamental" is not appropriate in the sentence, "Consequently, creating a version that was easy to understand appears to be fundamental." Maybe it should be changed to "important." And, "According to expert consultation" might be changed to "After consulting with experts..."

The words " positive contribute" do not make sense in the sentence ending with..... "meaning that that all items were related to each other and that positive contribute to measure the same general construct."

There are other grammatical problems in the paper- these are just come examples that I am pointing out as examples.

Author Response

Point 1. I think that a more detailed explanation of the CBR program and its strategy should be included in the Intro of the paper.  Response 1. Agreed, thank you. We modified introduction section. 

Point 2. There are some grammatical mistakes in the paper. For instance, in the second paragraph of the Discussion, the word "adapt" should be "adaptation of" in the sentence, "As a matter of fact, this step led to an improvement in the comprehensibility of the questions and adapt them to the specific Italian context." Also, the word "fundamental" is not appropriate in the sentence, "Consequently, creating a version that was easy to understand appears to be fundamental." Maybe it should be changed to "important." And, "According to expert consultation" might be changed to "After consulting with experts..."The words " positive contribute" do not make sense in the sentence ending with..... "meaning that that all items were related to each other and that positive contribute to measure the same general construct."There are other grammatical problems in the paper- these are just come examples that I am pointing out as examples.

Response 2. Thank you. We modified according to your suggestion and we performed new English editing to check grammatical mistakes in the paper

Reviewer 3 Report

This article refers to the validation of the Italian version of a questionnaire.

The authors did not follow the editorial indications, in particular for the abstract, which is structured, and in the indication of the references.

The abstract should not contain complete sentences of the text, such as “CBR has long lacked a strong scientific evidence foundation, due in part to the absence of standardized indicators” that in the text it is a statement attributed to an external source. It is not easy to identify the article to which the authors attribute this statement, since the references are reported in an inappropriate way. It would seem to be: Evans PJ, Zinkin P, Harpham T, Chaudury G. Evaluation of medical rehabilitation in community-based rehabilitation. Soc Sci Med. 2001 Aug;53(3):333-48. doi: 10.1016/s0277-9536(00)00321-x. At the moment the term "Community based rehabilitation" on PubMed has over 18,500 results, with nearly 700 systematic reviews and meta-analyses, and it is difficult to believe that the claim is still valid. We must consider that this is an article from twenty years ago, which does not review all the material available at the time, and which is very old and outdated. Consequently, the statement is not proven and cannot be inserted in the text, much less in the abstract.

The paper begins with a brief history of Community Based Rehabilitation (CBR) but does not give a definition of the term. Instead, authors should start with a definition.

The second paragraph of the Introduction begins with a sentence whose meaning is not clear: “To date, the CBR strategy has been developed in more than 90 countries, however many people with disability still have low access to healthcare services and rehabilitation”. Do the authors think that the CBR strategy is effective, but not sufficiently implemented, or that it is not effective, how does the fact that there are access problems in 90 countries that still apply it imply?

Materials and Methods. The authors calculated the needed sample size. Did they test the power calculation of the sample? This aspect is important because, with a group of only 40 people with disabilities, the confidence interval is very large. Consequently, it cannot be accepted that the group of cases is homogeneous in age with the controls only because the very large confidence interval includes both age values, 38 and 42 years. The two groups, cases and controls, probably differ in other factors, such as the level of education, which has not been tested, but which could be higher in the controls who are predominantly university students and teachers. This is a serious limitation of the study.

The authors declare that they have introduced changes in the questionnaire, questions H06 and H09. After these changes, they should have performed a confirmatory factor analysis (CFA) to verify the correspondence of the changes with the original, testing whether measures of a construct are consistent with the nature of that construct.

However, authors should conduct at least one exploratory factor analysis (EFA) to verify whether the structure of the original questionnaire has been preserved in the Italian translation. This is indispensable precisely because of the difficulties encountered in the translation, which the authors account for in the discussion and which could have changed the meaning of the questions and especially the way in which they are grouped into factors.

The statistical analysis is incomplete and does not allow to support the claims that the authors make about the validity of the Italian version of the tool.

Claims on the effectiveness of the tool used are not substantiated by scientific data. The statistical analysis is incomplete and does not allow to support the claims that the authors make about the validity of the Italian version of the tool.

Claims on the effectiveness of the tool used are not substantiated by scientific data. The purpose of CBRi is not to make a difference between healthy and sick, but to study the integration of the handicapped into the community. It, therefore, makes little sense to compare cases from various parts of Italy with students who have no relationship with them.

Author Response

Point 1: the abstract should not contain complete sentences of the text, such as “CBR has long lacked a strong scientific evidence foundation, due in part to the absence of standardized indicators” that in the text it is a statement attributed to an external source. It is not easy to identify the article to which the authors attribute this statement, since the references are reported in an inappropriate way. It would seem to be: Evans PJ, Zinkin P, Harpham T, Chaudury G. Evaluation of medical rehabilitation in community-based rehabilitation. Soc Sci Med. 2001 Aug;53(3):333-48. doi: 10.1016/s0277-9536(00)00321-x. At the moment the term "Community based rehabilitation" on PubMed has over 18,500 results, with nearly 700 systematic reviews and meta-analyses, and it is difficult to believe that the claim is still valid. We must consider that this is an article from twenty years ago, which does not review all the material available at the time, and which is very old and outdated. Consequently, the statement is not proven and cannot be inserted in the text, much less in the abstract.

Response 1: Thank you for your comments and suggestions. We modified abstract and references. We also integrated new evidences in the main text for supporting our research.

Point 2: The paper begins with a brief history of Community Based Rehabilitation (CBR) but does not give a definition of the term. Instead, authors should start with a definition
Response 2: Thank you. We agreed with you. Therefore we added the definition provided by the World Health Organization: "Community-based rehabilitation (CBR) is community action to ensure that people with disabilities have the same rights and opportunities as all other community members"

Point 3: The second paragraph of the Introduction begins with a sentence whose meaning is not clear: “To date, the CBR strategy has been developed in more than 90 countries, however many people with disability still have low access to healthcare services and rehabilitation”. Do the authors think that the CBR strategy is effective, but not sufficiently implemented, or that it is not effective, how does the fact that there are access problems in 90 countries that still apply it imply?

Response 3: We agreed, probably the sentence was confusing, We modified it.

Point 4: The authors calculated the needed sample size. Did they test the power calculation of the sample? This aspect is important because, with a group of only 40 people with disabilities, the confidence interval is very large

Response 4: As reported in methods, for psychometric properties we followed COSMIN methodology for defining adequate sample. Therefore a minimum sample size of 200 people was considered adequate. As reported in limitations sub-section, the study provides prelimianry evidence for using CBR-Is in Italy. We specify more in depth in limitations.

Point 5 : Consequently, it cannot be accepted that the group of cases is homogeneous in age with the controls only because the very large confidence interval includes both age values, 38 and 42 years. The two groups, cases and controls, probably differ in other factors, such as the level of education, which has not been tested, but which could be higher in the controls who are predominantly university students and teachers. This is a serious limitation of the study.

Response 5: Thank you for your comment. It is true that the sample size is limited, in fact we specify in the title and pointed in limitation that the paper reports preliminary evidence on disability. Therefore, we agreed with you and we specifiy it more in depth in limitiation. However, despite small sample size, sample is homogeneous for age and gender. We explore differences in Education of people with and without disability as results. (For response on university students please see Response 8)

Point 6: The authors declare that they have introduced changes in the questionnaire, questions H06 and H09. After these changes, they should have performed a confirmatory factor analysis (CFA) to verify the correspondence of the changes with the original, testing whether measures of a construct are consistent with the nature of that construct. However, authors should conduct at least one exploratory factor analysis (EFA) to verify whether the structure of the original questionnaire has been preserved in the Italian translation. This is indispensable precisely because of the difficulties encountered in the translation, which the authors account for in the discussion and which could have changed the meaning of the questions and especially the way in which they are grouped into factors.

Response 6: Thank you for your comments. The CBR Indicators have a theoretical construct that reflect the five components of the CBR matrix. To the best of our knowledge EFA should be performed when we have no a priori hypothesis about factors or patterns of measured variables. Furthermore, as explained in the Discussion, the difficulty in translation regards on semantic domain not on factor structure or item-reduction. Therefore, we did not perform a CFA. 

Point 7:  The statistical analysis is incomplete and does not allow to support the claims that the authors make about the validity of the Italian version of the tool.Claims on the effectiveness of the tool used are not substantiated by scientific data. The statistical analysis is incomplete and does not allow to support the claims that the authors make about the validity of the Italian version of the tool.

Response 7: Thank you for your comments. For the present study, we followed the COnsensus-based Standards for the selection of health status Measurement INstruments. The COSMIN suggests using cronbach alpha and and intraclass correlation coefficient for reliability. Therefore our analysis seems to be valid for internal consistency and intra-rater reliability. However, we agree woth you: using the word "valid" it seems not appropriate. Therefore, we opted to use "consistent" and "reliable" 

Point 8: Claims on the effectiveness of the tool used are not substantiated by scientific data. The purpose of CBRi is not to make a difference between healthy and sick, but to study the integration of the handicapped into the community. It, therefore, makes little sense to compare cases from various parts of Italy with students who have no relationship with them.

Response 8: Thank you for your comment. However, we did not perform and did not analyse effectiveness of CBR-Is. We performed a T-student for independent sample for measuring differences between two groups. This is the focus of the CBR indicators, as reported in the manual of the World Health Organization. For example the manual reports; "In this case the survey needs to be conducted in the community where CBR is being implemented and include people with disability and people without disability" In the manual, authors also reported different analysis comparing results for people with and without disability. This difference can be expressed in percentage terms or, as we did, through a comparison of the mean scores for those quantitative variables. At the end, it is important to point out that among the few students who participated in the study (few because the mean age of the Italian university students is 22 years-old, while the mean age reported is substantially higher) there were students with disabilities. So we think that it makes sense for the research purpose.

Round 2

Reviewer 3 Report

The authors responded to theThe authors only partially replied to the observations that were made. They improved the abstract and introduction, but did not respond to objections to the statistics used. The very limited time that the journal allows for review prevents you from carrying out the indicated analyzes. Authors should have more time to statistically improve their work before publishing it. best of their ability to the observations that wer

Author Response

criticisms have been made about the statistical analysis, but we are confident that we have acted according to international guidelines and recommendations. In fact, for the evaluation of psychometric properties, we followed the COnsensus-based Standards for the selection of health status Measurement INstruments (please see https://www.cosmin.nl/). We hope the following explanation can clarify our methodology.

Point 1: EFA and CFA

Response 1: A factor analysis could not be performed for the following reasons: The CBR Indicators have a theoretical construct that reflect the five components of the CBR matrix. To the best of our knowledge EFA should be performed when we have no a priori hypothesis about factors or patterns of measured variables, while CBR indicators are based on 5 factors defined by the 5 domains of the CBR matrix. Furthermore, as explained in the Discussion, the difficulty in translation regards only semantic domain not on factor structure or item-reduction. Therefore, we did not perform a CFA.

Furthermore, to perform EFA we should have answers that have the same construct. Some indicator responses, however, are purely qualitative and simply explain the reasons why people with and without disabilities fail to access rehabilitation or health services. The scoring system does not have the same "weight", in others they do not have a real score (the responses are qualitative) so it is not possible to perform an EFA. As anticipated, the basic theoretical construct instead is based on the domains of the CBR matrix so we referred to these domains.

Probably, we can perform a EFA considering only some items of the CBR-Is, but we think this make no sense.

Point 2:  The statistical analysis is incomplete and does not allow to support the claims. 

Response 2: To measure psychometric properties of a patient-reported outcome measure, we followed the COnsensus-based Standards for the selection of health status Measurement INstruments. For reliability, the COSMIN suggests using cronbach alpha and and intraclass correlation coefficient. Please see COSMIN Taxonomy of Measurement Properties:

https://www.cosmin.nl/tools/cosmin-taxonomy-measurement-properties/

As you can see, other recommended analysis was the use of Omega Coefficient. However, Omega can be performed when the covariance among the items can be approximately accounted for by a one-factor model. Based on the theroetical construct of the CBR-Is (five factors) we performed only the cronbach alpha for internal consistency. For intrerate reiliability, we performed ICC, as recommended by the COSMIN.

We believe that the analysis to investigate internal consistency and reliability were appropriate. We specified to follow COSMIN into the manuscript

Point 3: The purpose of CBRi is not to make a difference between healthy and sick, but to study the integration of the handicapped into the community. It, therefore, makes little sense to compare cases from various parts of Italy with students who have no relationship with them.

Response 3: As reported by the WHO, one of the focus of the CBR indicators, was to measure differences between people with and without disability. Please see the manual:

In the manual, WHO reported different analysis comparing results for people with and without disability. This difference can be expressed in percentage terms or, as we did, through a comparison of the mean scores for those quantitative variables. We performed an independent sample T-Test to compare mean scores between groups.

However, as suggested into previous comments we specified in "limitation section" that data cannot allow to generalize our finding to the whole Italian population.

At the end, for sampling we already answered that students with and without disability were included, as well as for other young or older people with and without. Furthermore, we think that having a sample with different geographical distribution provide more strenght to the study.